

# Taking the pulse of nature
## – How robotics and sensors assist in lake and reservoir management

**Sebastian Zug[1*†], Gero Licht[1], Erik Börner[2], Edjair de Souza Mota[4], Roberval Monteiro Bezerra de Lima[3,a],**
**Eric Roeder[2], Jörg Matschullat[2, 5*†]**
[1]Institute of Computer Science, Technical University Bergakademie Freiberg, Bernhard-von-Cotta-Straße 2, 09599
Freiberg, Germany
[2]Interdisciplinary Environmental Research Centre, Technical University Bergakademie Freiberg, Brennhausgasse 14,
09599 Freiberg, Germany
[3]Embrapa Florestas, Estrada da Ribeira, km 111, Guaraituba, Colombo-PR, 83411-000, Brazil
[a]formerly at: Embrapa Amazônia Ocidental in Manaus
[4]iComp, Universidade Federal de Amazonas, Av. Gen. Rodrigo Octávio, 6200 Setor Norte do Campus Universitário –
Coroado, Manaus-AM, 69080-900, Brazil
[5]Arthur L. Irving Institute, Dartmouth College, 33 Tuck Mall, Hanover, NH 03755, USA

12               [†]These authors contributed equally to this work and share first authorship

13        **Correspondence:** Sebastian Zug (robotics and informatics) and Jörg Matschullat (everything else):
14        Sebastian.Zug@informatik.tu-freiberg.de, matschul@tu-freiberg.de (joerg.matschullat@dartmouth.edu)

**Keywords:** Unmanned surface vehicle, robustness, autonomous field robot, autonomous data aggregation,
limnology, Amazon basin
**Abstract**
Ecosystems, like almost any environmental entity, are often highly sensitive to the presence of humans when
measuring field characteristics. Robotic solutions deserve attention to avoid or greatly reduce related bias.
Constant availability of robotic solutions, independent of the time of day and most weather conditions, is an
additional advantage.
Here, we present an autonomous, Modular Aquatic Robotic Platform (MARP-FG) designed to collect relevant
environmental information from surface waters. We define the demands, describe the encountered
obstacles and how to overcome them. MARP-FG implements autonomous navigation and data collection
capability across various floating-body configurations and sensor setups. Depending on the weight of the
measurement system (payload), catamaran floaters with a length ranging from 1.2 meters to 2.5 meters are
used. We realized and evaluated three different payloads based on the MARP-FG concept: i) Hydrographic
profiling with a multi-parameter probe, ii) Sonar-based 3D mapping of complex basins, and iii) Dynamic
closed chamber-based greenhouse gas exchange determination with on-board $CO_2$ quantification (IR
spectrometry) and gas sampling (Exetainers®) for subsequent gas-chromatographic analysis.
This work focuses on option iii) as a practical example to describe our design process and operational modes,
thus minimizing faults and errors, especially in harsh environments. Full operation was possible to wave
heights of ±40 cm and wind speeds to 7 m sec$^{-1}$. Positioning accuracy during measurement cycles was on
average better than ±2 m in xy directions. The platform has demonstrated its capabilities in field campaigns
on lakes in the Amazon basin (Brazil) and on waterbodies in temperate climate regions of Europe. Largely
improved and reproducible positioning on a waterbody, full functionality also under adverse weather
conditions and during nighttime significantly enhanced high-quality data acquisition and opens new
applications.
**1  Introduction**



Limnologists, biogeochemists and geoecologists in academia and water authorities strive to better
understand the responses of waterbodies to global change in order to support resilience and to maintain
biodiversity and aquatic ecosystem services at large. This requires robust methodological solutions.
Perpetual, careful and accurate observations are needed to provide reliable data and to secure high-quality
water for the public and for ecosystems. Tasks on surface waterbodies include regular water-column
sampling and hydrographic profiling (physicochemical parameter acquisition from surface to bottom). This
may be done once a month at reservoirs, for example in Germany (ATT 2021), and about monthly for surface
waters in the US (Riskin et al. 2018), or quarterly as recommended by the European Water Framework
Directive (Ziemińska-Stolarska et al. 2019). Such frequencies may be suboptimal given possible risks of
intentional or accidental disturbances (spills, contamination, etc.). Marcé et al. (2016) present related
challenges and demands of reliable and trustworthy waterbody monitoring. Less frequent but important
tasks include 3D-imaging of basins and their sedimentary structures (Fang et al. 2023), high-resolution
assessment of greenhouse gas fluxes ($CO_2$, $CH_4$, $N_2O$) between a waterbody and the atmosphere (Huttunen
et al. 2003), and other specialized studies, such as the occurrence of micro- and nano plastics in the water
column (Strungaru et al. 2019; Triebskorn et al. 2019).
Today's standard monitoring cannot be carried out in difficult weather conditions (e.g. thunderstorms) or at
night because of the potential risk to the personnel involved. To minimize associated risks, to improve the
accuracy and precision of data collection, and to increase the frequency of observations, robotic monitoring
appears to be a solution (Dunbabin and Marques 2012). This is not fundamentally new; there are very
different approaches (e.g. Dunbabin and Grinham 2010; Hitz et al. 2014; INTCATCH 2023; Jeong et al. 2020;
Melo et al. 2019; Mendoza-Chok et al. 2022; Rajewicz et al. 2022). However, current solutions tend to be
heavy, bulky, and costly, resulting in limited flexibility and versatility.
Based on the development of a versatile and robust closed dynamic chamber system for terrestrial
applications (Oertel et al. 2016; Pape et al. 2009; Rochette et al. 1997), we designed, built, and tested the
custom-built catamaran-body-based autonomous Modular Aquatic Robotic Platform (MARP-FG; Fig. 1). We
deployed this robotic platform to remote freshwater lakes in the state of Amazonas, Brazil, to investigate
their role in the global carbon cycle (Matschullat et al. 2024). The choice of the Amazon basin as
experimental region was technically motivated by the experience that equipment that withstands its harsh
climatological conditions (high temperatures with very high radiation and humidity) is robust enough to be
used anywhere in the world. Our intention was to develop a platform that can a) be transported easily,
b) quickly switch between different payloads, c) operate safely even under harsh conditions, including
d) nighttime operation, which may be too risky for human presence. The team repeated this investigation in
a multi-year campaign (project RoBiMo-Trop), continuously improving the robots in the process. Our main
research questions (RQ) for the development of our platform and related hypotheses (H) were:
RQ 1    Which general environmental conditions must be considered for a swimming robot operating in the
Amazon region? What specific requirements for the measurement process need to be considered
from a scientific perspective when designing the robot and its sensory system?

H1: Tropical weather conditions are the biggest source of error in implementing an autonomous
measurement system.

RQ 2    How small (and light) can a floating platform be to remain easily transportable and still always
function reliably under typical conditions of freshwater bodies in all climate zones?

H2: A catamaran shape can be small enough to fit mounted onto a standard pickup truck, light
enough to be accepted in international air traffic, and robust enough to withstand extreme
temperatures and humidities.

RQ 3    Which hardware/software architecture provides flexibility while maintaining testability and error
tolerance?



H3: A clear separation of tasks – here, data collection and autonomous navigation – ensures both the
expandability of the setup and the robustness of the overall system.

RQ 4   How to ensure that error states can be recognized as fast as possible?

H4: To detect and address fault conditions under harsh conditions, a multimodal approach is
required for error identification and communication. This approach should consider the situation's
specifics such as limited communication bandwidth, necessary information content, and intervention
capabilities.

## 2  Methods


Earlier developments implemented a platform for mapping water bathymetry using multibeam sonar, as well
as a platform transporting a winch-based system for multi-sensor probe determination of hydrographic
profiles (Fig. 1). Its footprint covers approximately 2.5 m x 1.4 m. However, these systems are too large for
convenient transport. Consequently, a redesign of the autonomous platform was needed to flexibly work in
the Amazon Basin environment and quickly move between waterbodies. This section describes, starting from
general requirements and local boundary conditions, the technical implementation and parallel hazard
analysis to ensure the required robustness.

### 2.1  General requirements and local conditions


The physical dimensions of the platform need a configuration that enables easy transport of the
disassembled MARP-FG easily by aircraft and in operational state by a standard pickup truck. A modular
structure must be realized, allowing for uncomplicated on-site assembly and quick payload exchange. The
robot should be as unobtrusive and quiet as possible for its environmental work and have minimum draught
to maneuver over very shallow terrain (≥ 20 cm).
The platform should not exceed a maximum distance of 2.5–3 km from the base station to allow retrieving it
in case of malfunction by boat, even under unfavorable wind and current conditions. It would be desirable
for the robot to work for an entire workday with a single set of batteries. That will cover distances of up to 10
km and allows taking nine gas measurement cycles of 30–45 minutes each. Frequent battery replacement at
the base station would be counterproductive, especially for long return distances to the measuring position.
At the start of the project, the team defined the goal for the robot to also work under night-time conditions.
Air temperatures can exceed 40 degrees Celsius in the Amazon Basin. Under direct sunlight, surface
temperatures can rise to 60 degrees Celsius; water temperatures reach up to 35 degrees Celsius. Solar
radiation (ca. 17 MJ m$^2$ day$^{-1}$; Malhi et al. 2022) and air humidity (77% in the dry season, 88% in the rainy
season; Met Office 2024) are mostly very high. Severe weather with tropical storm precipitation and strong
winds is common. Road conditions can be bad, exerting strong mechanical stress on any construction and to
electronics during transport. The investigated lakes showed water depth from < 0.5 to > 30 m (Matschullat et
al. 2024).
Mobile phone or WLAN connections are limited in the operating area. Actual data and state information can
reliably be exchanged between robot and devices (base station, monitoring tablets) in a local network only.
Its range is limited without complex additional antenna technology. As some of the measurements take place
in cultivated landscapes, the robot will 'encounter people'. The platform should therefore be visible as a
research platform with environmental tasks. In addition, every opportunity should be taken to explain the
idea and the functional principle of robot and project to interested parties. The team was aware, however
that the robot could not be protected in the event of a physical attack (which we never experienced).
The determination of gas exchange (here $CO_2$, $CH_4$ and $N_2O$) between waterbody and atmosphere requires a
fast on-board spectrometer for direct $CO_2$ quantification, and the ability to sample gases for subsequent gas-
chromatographic analysis in a laboratory. Parallel to gas analysis and sampling, ambient parameters (water
temperature, air temperature, pressure and humidity, photosynthetically active radiation-PAR, and wind
speed) must be registered in high temporal resolution to be able to evaluate the obtained gas data. When



measuring gas exchange, the robot should keep its position stable within a range of ±3 m regardless of wind
and wave dynamics (see 2.5).
**2.2    Platform and payload design**
To meet the handling and campaign requirements – maximum size and payload, integrated multimodal
sensors (Table 1), and platform stability issues – the team decided to develop a custom design in our
workshop. Two fiberglass floats (each approximately 120 x 20 x 20 cm, LxWxH) connected by a universal
aluminum frame (4 x 4 cm) form the basis of the easily dismountable MARP-FG catamaran platform (Fig. 2).
Quick-release fasteners allow for quick exchange between payloads.
Depending on the payload, the total weight is between 20 kg (mounted platform) and up to 50 kg (with
sonar). Here, we focus on the 'chamber system' configuration (ca. 32 kg; Fig. 2, Table 1). The vertical and
straight interior sides of the floaters, in combination with deflectors mounted onto the movable chamber,
provide perfectly still water conditions inside the chamber during measurements, independent of wind
shear, waves, etc. – perfect for undisturbed gas exchange determinations. Material costs for the platform,
including thrusters, batteries and steering unit, were about 3,000 €. The chamber system with all associated
sensors added another 3,500 €, including the gas sampling unit. The bridge with micrometeorological sensors
was about €1,500, for a total of €7,500 for a fully functional system (plus land-based receivers/laptop
computers; prices from summer 2024).
The design strikes a balance between manageable size and stable movement and positioning on water, even
under more challenging weather, wind and current conditions. Two centrally mounted electric thrusters (625
W each; T200, Blue Robotics Inc., USA) power the boat and allow for top speeds of 5–6 km h$^{-1}$ with the
chamber system as payload. These thrusters can be fine-tuned to keep the platform in place with position
stability of ± 1–2 m$^2$ (Fig. 6). With thrusters and payload, a draught of 15 cm is realized, allowing to cruise
across rather shallow waters, too. Four rechargeable batteries (5.2 Ah, 18 V, Einhell, Germany) provide up to
8 hours of system operation. The battery management system supports hot swapping (battery change while
the application Is running), which makes the system more flexible. A high-resolution (accuracy ± 1 cm) sensor
(Ping Sonar Altimeter and Echosounder, Blue Robotics Inc., USA) permanently records water depth (Table 1).
Box 1 (yellow at the stern in Fig. 2) contains the power supply, a high-resolution GNSS receiver, and an
inertial measurement unit (accelerometer, gyroscope, compass) to collect additional navigation information.
The gray box at the bow is the actual control unit of the chamber system. It houses a web server for
intermediate access to the current state of data aggregation (Fig. 2). In addition to position information and
gas concentration, the robot records wind speed and direction, water depth, temperature, humidity, and
ambient light (PAR) information, necessary for evaluating gas exchange data. Box 2 (yellow in Fig. 2) contains
the xy-driven, 10/20 mL syringe-based gas sampling unit that draws gas from the chamber to feed 18
Exetainer® flasks at discrete times (Fig. 4). The unit enables three gas sampling sequences for subsequent
analysis of $CO_2$ and other gases. The sampling unit is connected directly to the chamber via valve-controlled
silicone tubing (not shown in Figs. 2 and 4).
The platform takes off from shore and autonomously navigates to its pre-defined position(s), like aerial
drones. Course and experimental sequence are pre-programmed at the base station
(https://ardupilot.org/planner/). The human pilot is still responsible for ensuring that the route is navigable
and free of obstacles. The robot then automatically executes the planned route and the assigned
measurement processes, including autonomous navigation and continuous monitoring of the actual
measurement process (Fig. 3). When a task is completed, the platform can move to the next position and
restart working. At the end of a site measurement, the platform returns to its starting position (Fig. 3b). In
case of error, the mission is aborted, and the robot returns to the base station. Throughout the mission, live
data transmission allows monitoring of the measurement and navigation processes (Fig. 3a). A radio link
between the robot and the remote control, as well as between the robot and a laptop or smartphone,
supports manual intervention.



The hardware/software architecture of the system is divided into two components: i) navigation and ii) the
actual measurement unit. Their strict separation simplifies the development process, decouples the systems
in case of failure, and ensures fast adaptation to new measurement tasks/sensor setups. Figure 3a illustrates
the basic structure and interactions. Autonomous navigation based on a commercial Pixhawk controller
(STM32 controller inside) is implemented (left side), an ESP32 controller realizes measurements and data
aggregation (right side). The Pixhawk controller is widely used in aerial drone applications and integrates
open software/hardware implementations at different levels of the autonomous navigation process. In our
setup, the Pixhawk runs a customized version of the Ardupilot software stack. The ESP32 implements the
actual $CO_2$ measurements and records environmental parameters (water temperature and depth, PAR and
micrometeorology). This part can be changed to another setup for alternative missions (modular design). The
communication interfaces of both microcontrollers can be addressed wirelessly by the pilot. The Pixhawk
provides a standardized telemetry interface to interact with the base station using the 'MAVLink' standard,
which works up to about 500 meters over water. A web server is also run on the ESP to display live
measurement status on a cell phone or laptop browser. This means that the supervisor has access to all
parameters (autonomous navigation, current data from the gas bell) and can intervene, if necessary, at least
over short distances.
**2.3   Application in a pilot study in humid tropical freshwater environments**
Five lakes in the state of Amazonas, Brazil, were selected for five field campaigns, covering two wet seasons
and three dry seasons from September 2021 to August 2023 (Matschullat et al. 2024). Artificial Balbina
reservoir, located about 180 km north of Manaus, the capital of Amazonas State, Brazil, is a clearwater lake,
filled in 1984. Blackwater lakes of the Negro River water type were represented by the Caldeirão and Jandira
lakes on the Iranduba Peninsula (between the Negro and Solimões Rivers). Lakes Iranduba and Grande
represented whitewater lakes of the Solimões (Amazonas) River water type. The project website shows the
locations of the lakes on a map (https://sebastianzug.github.io/RoBiMo_Trop_DataSet/; Zug 2023).
**2.4      $CO_2$ exchange, analytical methods and boundary conditions**
To determine $CO_2$ exchange, a closed dynamic chamber system was mounted to the upper quick-release
aluminum frame. The custom-built chamber automatically rises above the water for flushing between
measurements and for safe travel and transport. To record a measurement, the chamber tilts down. Its base
then sits 3–4 centimeters below the water surface to prevent atmospheric air from being drawn in. An
infrared spectrometer (GMP-252, Vaisala, Finland) takes high-resolution $CO_2$ measurements at 1-second
intervals. The two deflector shields mounted on the chamber between the two floats drastically reduce wave
motion and currents around the chamber (Fig. 2). A fan inside the chamber provides gas homogenization
during the accumulation period.
At each position, the measurement sequence consists of three repetitions (approximately 6 minutes each) of
$CO_2$ determinations with intermittent purging. A fourth repetition starts an automatic parallel gas sampling
series. Each series consisted of six samples taken at equal time intervals (approximately 30 minutes total)
and stored in double septum 12 mL Exetainer® flasks (Labco, England). At the end of a series, the chamber
tilts up for flushing, and the platform can move to the next predefined position. All sensor parameters are
permanently logged. After drying steps in the laboratory, the samples from the Exetainers were analyzed for
$CO_2$ plus methane ($CH_4$) and nitrous oxide ($N_2O$) by gas chromatography (SRI Instruments 8610C, USA) in
Freiberg under thorough quality control.
**3      Mechanisms to increase robustness**
Field research with technical equipment often faces difficulties when parts of the system malfunction.
Limited repair options in the field and a potential lack of communication channels in remote regions can
significantly hinder or even cause a measurement campaign to fail. Unforeseeable errors can be categorized
according to the different phases of a mission (Table 2, columns 1 and 2). Based on this categorization, the



team applied a hazard analysis during the design process of the MARP-FG to identify potential sources of
error, considering the system architecture, usage conditions, and technical components. This error list was
continuously updated with each campaign's experience. The third column in Table 2 provides examples of
individual error sources. Each error was mapped on three types of strategies (S):
S1. Incorporating improvements into the design to eliminate the errors,
S2. Developing an operational avoidance strategy to minimize the likelihood of error occurrence, or
S3. Including the error in the project's monitoring concept, making the error status explicit and visible to
the observer.

The continuous improvement of the technical design led to protective covers for prominent sensors (Table 2:
1a), extensive routing of cables in appropriate channels, and the introduction of two pendulum flaps to
stabilize the water between the catamaran floats (Table 2: 6d). An example of the avoidance strategy is the
procedure for aggregating and handling the data collected by the robot (Table 2: 7cd). After each mission,
the team had to minimize the chances of data loss. A detailed procedure model was designed describing the
processes for removing the SD card from the navigation unit and reading out the data from the measuring
gas bell. Accordingly, the processes initially prescribed the shutdown of the entire system, a ban on removing
the storage media near water, and an immediate multiple copy operation for at least two independent
storage devices. At least one of these copies had to be checked for file consistency. Data loss was therefore
ruled out at this stage of the project.
However, it was impossible to find a suitable avoidance strategy for all potential errors. In particular, the
complex software structure with the two sub-areas of autonomous navigation and control of the
measurement system and their interaction opened the possibility of software errors (Table 2: 4ab). The aim
here was to ensure early detection so that the fault could be rectified immediately in the field by repairing or
restarting the system. The diversity of faults, the need to display the fault status over different distances and
the challenges of the application clarified that no standardized interface for communicating the faults makes
sense. Accordingly, various, partly redundant channels were set up (Table 3 in descending order of spatial
range).
The composition of the different error communication channels reflects the specific requirements of the
RoBiMo-Trop campaign. The robot should operate on waterbodies during the day and at night. For safety
reasons, the robot should not be accompanied by a boat in the dark. Hence, communication of the robot's
states had to be provided up to a maximum of 2.5 km, without 3G to 5G mobile phone connection. Direct
radio communication with the telemetry unit is not robustly possible over open water surfaces at ranges of
more than 500 m. Following these boundary conditions, three patterns of online-mission monitoring were
implemented in the field tests that can be considered successful:
1) During the first missions, the team members accompanied the robot closely with a small boat during
259         day missions. Thus, continuous visual and data-driven monitoring ensured that errors could be
detected quickly (Table 2 error classes 1, 2, and 3).

2) During nighttime journeys, initial measurements were carried out only in the immediate vicinity. In the
range of the telemetry, the correct behavior of the robot was tested with a few measuring points.
Thereafter, the MARP-FG operated autonomously at the maximum distance.

3) Due to the limited bandwidth and range of wireless communication, a complete online evaluation of
the whole data set is impossible. Hence, it was necessary to check the results offline after each mission.
Table 3 highlights these methods in gray.

The following paragraphs describe the implementation of the four channels mentioned in Table 3 as
strategies for error communication: Position lights and LED beams, webserver for state representation,
automated gas sampling unit and the data processing chain.





**Position Lights and LED Beams.** The position lights and LED lighting enable the robot to be localized during
night missions and to visualize its status. Since the initial LED strip was not bright enough to display the
position over 2500 meters, the team integrated position lights. These simple solar-powered LED lamps were
mounted on the bridge next to the antennas on the left and right. Their luminosity was so strong that the
illumination was also very useful for preparing the platform and for manual navigation, considering the
vegetation along the shore.
The LED strip with a WS2812B chipset could not perform the intended task. It was supposed to visualize
different phases of the mission implementation (see Figure 2b) and communicate navigation-specific errors
such as low battery level, increased current consumption of the motors, and missing GNSS position
information. Due to time constraints, the state-dependent color selection from the PixHawk has not yet been
implemented. The team integrated an additional microcontroller responsible for initializing and controlling
the LEDs. However, these LEDs then displayed a static pattern that indicated the direction of the robot based
on the colors.
**Webserver.** The web server (Figure 5) provides direct access to the parameters of the gas chamber
measurement process and currently recorded values. This includes supplementary sensors for measuring
environmental conditions. The functionality was implemented as a task within the FreeRTOS-based software
structure of the ESP32. For performance reasons, it was assumed that only one client would be connected at
a time. Due to the assumed lack of internet connectivity in the operational area, all necessary JavaScript
libraries used for graphical representation were stored locally on the ESP. This allowed the client's browser
to retrieve them directly from the main webpage. Using a tablet or mobile phone for this interface proved
effective. However, despite an external antenna for the ESP, the communication range was very limited.
Stable communication was only possible up to 20 meters.
**Gas sampler.** Parallel to the continuous measurement of the gas composition in the chamber during
measurements, the MARP-FG activates a sampler that transfers gas from the chamber headspace into
Exetainers®. The redundant data collection enables to check the plausibility of the digitally recorded values
afterwards and to evaluate other gases.
The gas sampler itself consists of a pump system with a capacity of 10 mL per stroke (Fig. 4). Before the
actual sampling, the system first flushes all lines with air from the chamber. The sampler then moves the
needle to the position of the next free Exetainer® and inserts it into the vacuumed glass tube, which is sealed
with a rubber membrane. The setup comprises a total of 6x3 pre-labeled Exetainers®. The sampler integrates
several error identification methods that are designed to detect if, for example, the needle has become
stuck, the needle positioning does not reference an Exetainer® flask or that a flask was not completely filled.
**Log file analysis.** The measurement and the robot control system store the collected data on individual
memory cards. The Pixhawk records all robot-specific information (steering commands, control states,
navigation parameters, internal robot states, battery system data) in standardized Ardupilot mission files; the
ESP32 logs the measurement data in CSV format (Fig. 3). The Ardupilot log files contain all configuration
parameters of the navigation unit and tracks of measurements/robot states with individual sampling rates
(https://ardupilot.org/copter/docs/logmessages.html). These logging parameters are highly parameterizable
(https://ardupilot.org/copter/docs/common-downloading-and-analyzing-data-logs-in-mission-
planner.html#common-downloading-and-analyzing-data-logs-in-mission-planner). Given individual problems
with radio outages due to long ranges and limited bandwidth, we did not consider the telemetry data set for
remote transfer, the second log chain available for Ardupilot. Accordingly, these logs do not cover the entire
mission records.
All aggregated information is processed offline in a Python-based toolchain
(https://sebastianzug.github.io/RoBiMo_Trop_DataSet/; Zug 2023). The implementation merges robot state
information from the Ardupilot log files and the measurement data in five steps:



1. The raw data aggregator evaluates and homogenizes the individual files. It maps the content on Pandas' data frames (Pandas: RRID:SCR_018214). The binary log files were transferred on text files in a first step and searched for specific log samples (estimated position, AHR2, gnss position measurements, GPS, and sonar outputs, RFND_Dist) by a python script at a second stage. For this purpose, an existing open source implementation was adapted for reliable search operations on log files ([https://gitlab.rrz.uni-hamburg.de/bay2789/bslogfiles/-/tree/master](https://gitlab.rrz.uni-hamburg.de/bay2789/bslogfiles/-/tree/master)).

2. The position and measurement data were merged using the time stamps. The necessary synchronization took place on the Pixhawk via the GNSS measurements while the ESP obtained a time stamp once at the beginning of the measurement via the connection with a mobile device with an active internet connection.

3. Cluster analysis was implemented to investigate the movement behavior of the robot in the vicinity of the manually selected measurement positions. Based on the dwell time at these points, the spatial position was extracted using a k-means approach. The number of clusters varied per water body. At the same stage, corresponding statistical key figures (min, max, std) of the water parameters ($CO_2$, temperature, depth, etc.) are summarized.

4. The visualization includes the georeferenced representation of the robot movements for the individual measuring points (Figures 3, 6). The script generated both an overall overview and a measurement point-related representation of the autonomous robot's movements. For the analysis of the robot behavior, position fidelity was important. Figure 6 shows the distributions and indicates maximum horizontal (2.6 m) and vertical (4.0 m) deviations. However, the histogram clearly shows that most of the measuring points were much closer to the intended position. Based on the analyses, adjustments could be made to the control parameters.

5. Step 5 automatically generates the web pages containing graphics and data. This concerns both the representation of the robot's movements (exemplary [https://sebastianzug.github.io/RoBiMo_Trop_DataSet/html/balbina.html](https://sebastianzug.github.io/RoBiMo_Trop_DataSet/html/balbina.html)) and the tabular processing of the measurement results for immediate evaluation and the planning of further missions on a body of water ([https://sebastianzug.github.io/RoBiMo_Trop_DataSet/html/interactive_table.html](https://sebastianzug.github.io/RoBiMo_Trop_DataSet/html/interactive_table.html)).

The software was realized as a collection of Jupyter notebooks, executed as a pipeline based on the papermill python package ([https://github.com/nteract/papermill](https://github.com/nteract/papermill)) when a new data set is available as a public GitHub project (Jupyter Notebook: RRID:SCR_018315; GitHub: RRID:SCR_002630). With this implementation, the data from MARP-FG can be analyzed daily in parallel with the measurement campaign and visualized on a website. This enabled the team to constantly review their measurement strategy and adapt the setup.

## 4    Results and discussion

While most of this work relates to the methodological development of the platform and its payload optimization, the next two sections will present key results of that effort and discuss them with respect to our initial research questions and hypotheses.

### 4.1    Robotics and Informatics

Starting with the first prototype in 2020, an all-manual catamaran and an early version of the chamber system, our robotic platform is currently in its 4th development stage (Fig. 7). Each stage added sensors and capabilities. With each step, the platform became more flexible and autonomous, leading to its current state of full autonomy. Larger antennas proved very useful in extending the range of the vehicle when direct communication during a run was required. Two information pipelines (Figure 3) were developed to evaluate errors, inaccuracies, and malfunctions.

The entire construction, including the gas exchange payload (mounted on a second, upper aluminum frame), works well up to wave heights of about ±40 cm and wind speeds up to about 7 m s$^{-1}$. Beyond these values,



the platform can still maneuver but does not perform reliable gas exchange measurements, 3D mapping or
hydrographic profiling.
The catamaran-type platform body presents maximum stability under most weather conditions and allows
for comparatively heavy payloads. The mid-board thruster positioning proved to be better than conventional
stern positioning, since it allows for minimum turning space and operation in demanding environments, such
as water surfaces with thick carpets of macroalga or other plant and drift materials. Figure 5 shows an
example of a real mission at Lake Caldeirão, Amazonas, Brazil. It shows that position 2 was targeted four
times by MARP-FG. The maximum difference in horizontal (x) and vertical (y) directions was 2.6 and 4.0 m,
respectively. The two histograms attached to the trajectories in x and y direction show that these maxima
were outliers.
All components resisted significant rain and windstorms; we did not face data losses. In more remote
environments with minimum nighttime luminosity, the additional mount of position lights on the bridge is
highly recommended. We successfully used Velcro-strap attached battery-powered LEDs.
The multimodal error communication concept has proven its worth. Regardless of whether
•    material got caught in the propellers,
•    the state machine of the measuring process blocked due to communication interruptions during
command transmission, or

•    the gas sampler blocked due to a shifted Exetainer® position,
all error states were quickly identified. The only points of criticism during the last campaign refer to missing
state information, represented by the LED bar, and the clarity of the position lights. The first aspect would
have eliminated the need for manual checks of the web server and telemetry data upon detecting an error.
Local error interpretation and visualization would have further enhanced convenience. The latter did not
allow for a clear indication of the robot's current direction of movement over long distances, especially
during the night missions, even with good visibility.
**4.2    Biogeochemistry – Geoecology – Limnology**
All these scientific fields contribute to Earth System Science and are particularly interested in freshwater
ecosystems and their role in the global carbon cycle (e.g., Friedlingstein et al. 2023; Raymond et al. 2013).
Our entire platform development (MARP-FG) was initially motivated by the necessity to produce high-quality
data at all times of day and under partly harsh meteorological conditions in the Amazon Basin, Brazil – with
limited funding available. The step-by-step development of hard and software proved helpful as it reduced
the complexity of errors and malfunctions – and allowed for successful field campaigns from the start.
The MARP-FG allows for work under forest canopy in the wet season, when water levels are high, and lakes
occupy significantly larger surface areas than during their low water conditions in the dry season. The slim
and height-limited construction could maneuver through thickets that are impossible to pass for people in a
small boat. The almost inaudible humming of the thrusters and the chamber mechanism and internal fan
noises do not deter animals. While large mammals such as Amazon river dolphins (*Inia geoffrensis*) curiously
investigated the platform, they never attacked or disturbed measurement and sampling. Even at nighttime,
the platform is non-audible as of about 10 m distance from the operators.
Greenhouse gas flux determinations require day- and nighttime data gathering (Oertel et al. 2016; Pape et al.
2009; Rochette et al. 1997). Related work on water bodies is still scarce; no homogenized methodology
exists. Solutions range from very simple makeshift floats carrying an equally simple chamber (both neither
robust nor able to work autonomously or at nighttime) to larger manned platforms with more sophisticated
chamber systems or eddy-covariance towers (Podgrajsek et al. 2014). The latter are expensive and
cumbersome to transport; certainly not to be taken along an aircraft with a research mission crew.



Starting out with a closed-dynamic chamber system, developed for soil gas exchange evaluations (Oertel et
al. 2016), we realized that this model was too large and heavy, and that the Vaisala sensor GMP 343 was
suboptimal with its higher energy demand. In addition, transparent chamber design was not necessary.
Given the obtained smaller dimensions and lighter construction, our redesigned chamber system could cope
better with the aquatic environment. The straight vertical board between the floaters was a necessity for the
payload 'chamber system' and related gas sampling to obtain truly quiet water conditions during
measurements.
When using the chamber system to measure $CO_2$ gas exchange on-site and to sample greenhouse gases, the
$CO_2$ data may vary (within tolerable limits) between the on-board determination by IRGA in comparison with
the samples taken with the automatic sampler. Such difference depends on the parameters 'air humidity',
'wind speed', and 'wave action', and can be explained by the Vaisala sensor sensitivity to water spray and
ambient humidity at large, while the automatically taken gas samples will be dried prior to gas-
chromatographical species determination, homogenizing them for subsequent analysis.
Achieved $CO_2$ determinations remained constant and highly reproducible throughout. Distinct day/night
differences in gas exchange could be confirmed as portrayed in the literature (Sieczko et al. 2020). Our new
data do not show significantly higher emissions of the waterbodies as compared with surrounding soils under
forest canopy or agricultural land use (Matschullat et al. 2021). A related manuscript is under preparation,
data will be uploaded to the Pangaea data publisher (https://www.pangaea.de/); interested readers are
welcome to contact the authors for more detail.

## 5     Conclusions

Robotic monitoring and sampling on water bodies can reduce bias and increase data accuracy and precision
due to improved position accuracy and reproducibility of tracks, as well as the absence of human
disturbance. Reduced risk to personnel and the ability to operate under more challenging conditions such as
overnight and during bad weather are additional benefits. The MARP-FG platform stays in position during
measurements and sampling with an average accuracy of 1–2 meters (X-Y-directions) and revisits predefined
positions with the same precision. The relatively low platform mass drastically reduced any unwanted
"pumping effect" of a water column compared to manned boats when making gas exchange determinations,
and its dimensions allow for cruising across both shallow water stretches and under overhanging tree or
brush canopy.
Measurements of the water column and surface parameters were spatially reproducible and enabled high-
resolution data, important to assess water quality in lakes with varying bottom morphometry, spatially
confined (underwater) inflow areas, and to enable specific experimental designs that require observation of
spatial phenomena in high temporal resolution.
Our initial platform development questions were answered as follows:
RQ 1     The floating autonomous robotic platform MARP-FG can be easily transported (standard pick-up
vehicle, air transport) and operate reliably under typical freshwater environmental conditions in all
climate zones, including harsh weather conditions. During the five campaigns in the Amazon Basin, as
well as numerous campaigns in Central Europe, the platform worked reliably in all weather
conditions, including strong storms and wave heights above ± 40 cm, as well as at night. With
charged backup batteries, 24-hour campaigns are possible.

RQ 2     Depending on payload, the MARP-FG weighs between 20 and 100 kg. With the 'chamber' payload,
the weight is 32 kg, and the outer dimensions 120 x 70 x 80 cm (LxWxH). The catamaran-type floater
design proved ideal for the defined tasks. Components with maximum reliability even under
challenging environmental conditions (e.g. sensors, thrusters, etc.) exist and are freely available on
the market,



RQ 3    A split coordinated architecture for mission control and payload functionality (data acquisition)
ensures smooth and reliable information and data transfer between the MARP-FG platform and a
ground station.
RQ 4    It is not enough to integrate diverse communication methods when operating a robot in the field.
Rather, these methods must be embedded within a comprehensive fault identification strategy that
ensures uncertainties with the platform can be reliably detected, if not avoided. This paper illustrates
the methodological approach on how to achieve this.

Especially for gas exchange measurements and hydrographic profiles, a robotic platform avoids errors caused
by the larger mass of a boat with people and its physical effect on the water column (pumping effect). Since
diurnal variability may be highly significant, our robotic approach allows for nighttime measurements and
sampling as well as daytime series of measurements. Our initial key questions have been answered, at least
for now, and will need to be verified in future campaigns.
**Conflict of Interest**
The authors declare that the research was conducted in the absence of any commercial or financial
relationships that could be construed as a potential conflict of interest.
**Author Contributions**
SZ: Conceptualization, Data curation, Investigation, Methodology, Resources, Software, Supervision,
Validation, Writing; GL: Data curation, Methodology, Software, Validation; EB: Conceptualization,
Methodology, Resources, Software; RMBL: Funding acquisition, Project administration, Resources,
Supervision; ER: Investigation, Validation; EM: Resources; JM: Conceptualization, Data curation, Formal
analysis, Funding acquisition, Investigation, Methodology, Project administration, Resources, Supervision,
Writing.
**Funding**
The EU-funded (ESF) project 'RoBiMo' laid the foundation for the development presented here, which was
funded by the German Federal Ministry of Education and Research (BMBF; FKZ 01DN21018) through its
International Bureau, and the Deutsche Bundesstiftung Umwelt (DBU; Az 36095/01).
**Acknowledgments**
This work is part of the project RoBiMo-Trop (Robotic Monitoring of Freshwaters in the Tropics), supported
by the German Federal Ministry of Education and Research and the Deutsche Bundesstiftung Umwelt. We
are grateful for their support, without which this project would not have been possible. It would also not
have been possible without the support of the ESF for the previous project, RoBiMo, which laid the
foundation for the hardware and software developments. A big thank you goes to Embrapa Amazônia
Ocidental in Manaus for providing the technical infrastructure and, at least as important, to Gilvan Coimbra
Martins (Embrapa) and Séan P.A. Adam (TUBAF) for their invaluable support in the field.

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

access: July 09, 2024.
**Data Availability Statement**
The datasets for this study can be found in the GitHub repository: [Zug S. Data storage and processing RoBiMo-Trop".
https://github.com/SebastianZug/RoBiMo_Trop_DataSet; 2023]; Last access: May 05, 2024.
**Tables and figures**



**Table 1.** MARP-FG payload sensor types and their specifics as well as producer information (all web pages last verified
on July 09, 2024)

| Sensor | Measuring range (accuracy) | Links |
|---|---|---|
| $CO_2$-infrared spectrometer GMP-252 | 0–2000 ppm$_v$ (±18 ppm$_v$). The sensor can be set to higher concentrations with lower resolution | https://www.vaisala.com/en/products/instruments-sensors-and-other-measurement-devices/instruments-industrial-measurements/gmp252 |
| Combined air humidity-temperature probe DKRF500 EA | 0–100 % RH (±1.8% RH)<br>-40 – +80 °C (±0.3°C) | https://www.driesen-kern.com/products/humidity-and-material-moisture/transmitters-and-probes/humidity-temperature-standard-model-dkrf500.php |
| Temperature probe DKT200 | -40 – +80 °C (±0.3°C) | https://www.driesen-kern.com/products/temperature-measurement/temperature-probes/dkt200-temperature-probe.php |
| Air pressure sensor AMS 4711-1200-B | 700–1200 mbar (0.3 % FSO) | https://www.amsys-sensor.com/products/pressure-sensor/ams4711-analog-pressure-transmitter-5v-output/ |
| PAR sensor Apogee SQ 421 | 1–4000 µmol m$^{-2}$ s$^{-1}$ (± 5 %) | https://www.apogeeinstruments.com/original-quantum-sensor-support/ |
| Anemometer ATMOS 22 | 0–30 m/s (±0.3 m/s)<br>0–359 ° (±1 °) | https://www.metergroup.com/en/meter-environment/products/atmos-22-ultrasonic-anemometer |
| Precip sensor RG-15 | (±10 %) | https://www.antratek.de/optical-rain-gauge-rg-15 |
| Air temp, rH, pressure sensor BME280 | -40 – +85 °C (±1.0 °C); 0–100 % RH (±3%); 300–1100 hPa (±1 hPa) | https://www.bosch-sensortec.com/products/environmental-sensors/humidity-sensors-bme280/ |
| Altimeter and echosounder Ping Sonar | 0.5–70 m (1–25 cm) | https://bluerobotics.com/store/sensors-sonars-cameras/sonar/ping-sonar-r2-rp/ |





**Table 2**. Potential error sources in different phases of work (A) with field robots. At a general level (B) and a specific
level for the described floating robot system MARP-FG (C)

| Phase (A) | Generic, project-/system overarching error sources (B) | Error sources in the RoBiMo-Trop project (C) |
|---|---|---|
| Mission preparation | (1) Carelessness when transporting the robot in the field | (1a) Damage to a permanently installed (weather) sensor, (1b) cabling on the robot, (1c) waterproof covers on the robot, (1d) the floats |
| | (2) Errors during mechanical/electrical system assembly in preparation for a mission | (2a) Incorrect thruster height settings before each use, (2b) Asymmetrical thruster mounting, (2c) Insertion of empty batteries, (2d) Failure to remove protective caps from sensors |
| | (3) Operating error during software-based mission preparation | (3a) Incorrect specification of the trajectory, (3b) Incorrect adjustment of the thruster control parameters |
| Mission execution | (4) Software error in the navigation unit or the measuring system | (4a) Crashes of individual components, (4b) Irregular timing of individual tasks |
| | (5) Hardware errors | (5a) Random disturbances of the sensor measurement processes, (5b) Jamming of the chamber during its movement |
| | (6) External disturbances | (6a) Strong currents, (6b) Obstacles below the water surface (leading to thruster blockage, (6c) Obstacles above the water (branches of trees), (6d) Strong wave movements during measurements |
| Mission evaluation | (7) Errors during data backup and preparation | (7a) Overwriting data, (7b) Errors in data recording, (7c) Loss of storage media, (7d) Loss of data records |




**Table 3.** MARP-FG robot error monitoring components. 'Range' summarizes the maximum distance to the supervisor, limited by visibility or range of wireless communication. 'Transferred data' specifies what can be transmitted via individual channels. The detectable 'error modes' with the information provided (numeration reference to Table 2).

| Item | Range [m] | Transferred data | Identifiable error modes (Table 2) |
|---|---|---|---|
| Position lights | ≤ 2500 (nighttime) | Position of the robot and its changes, (particularly during nighttime) | 3ab, 4a |
| Illuminated ring with 24 LEDs | ≤ 1000 (nighttime) | Abstract state modes such as "autonomous cruise" or "Measurement in progress" | 4ab (knowing the specified time periods) |
| Telemetry | ≤ 500 | Status of the robot, its position and related changes, water depth, battery status | 1ab, 2ab, 3ab, 6d |
| Webserver of the chamber system | ≤ 20 | Current data of the momentary measurement of the chamber | 4ab, 5a |
| Gas samples | 0 | Implicit data in the form of gas samples taken in parallel with the measurement | 5a |
| Chamber and MARP/FG log files | 0 | Complete overview of the mission and measurement data of a campaign | 3ab, 6d |

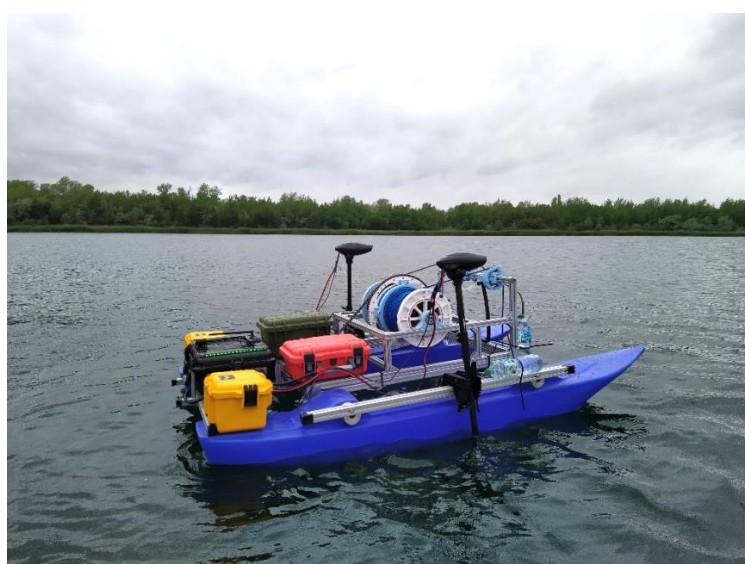

**Fig. 1**. An expanded implementation of the MARP-FG concept supports a payload capacity of 100 kg. This enables the platform to carry an ultrasound scanning system or, shown here, a winch with a multisensor probe capable of descending to depths of up to 70 m water depth. The thrusters are Minn Kota electric drives





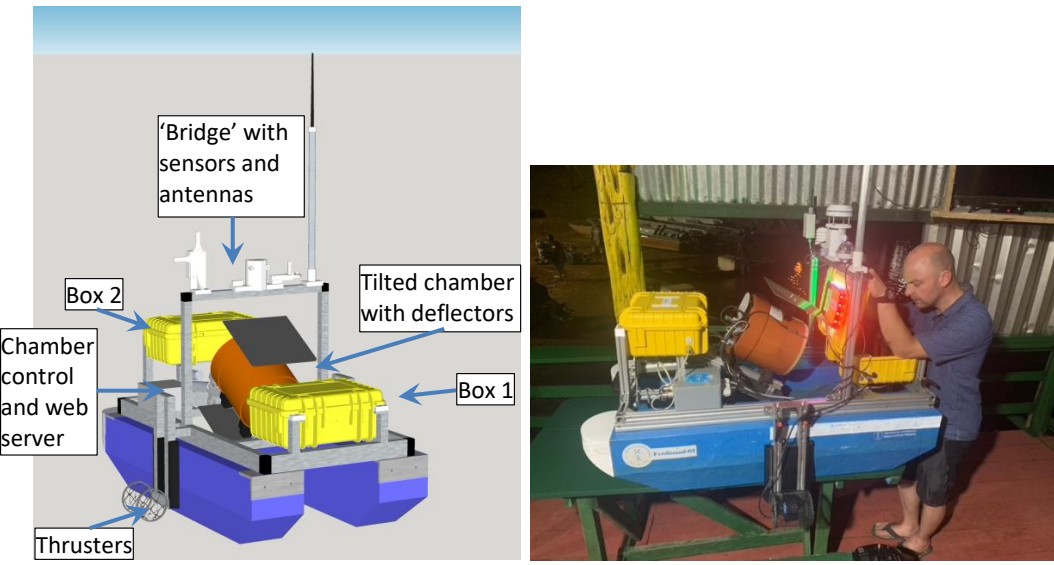

590

**Fig. 2. a)** MARP-FG with chamber system. On top, communication antennas, and micrometeorological sensors for wind,
humidity temperature and precipitation (white). Below the bridge, at rear, the chamber-tilting mechanism, and the
automatic gas sampling box 2 (yellow). Two thrusters are mounted at the center. The chamber (open during transport
and flushing) sits in the center inside (orange); deflector shields are visible (gray). In the front: Power supply and
positioning equipment in box 1 (yellow). **b)** The real MARP-FG prior to a launch. See text for more detail

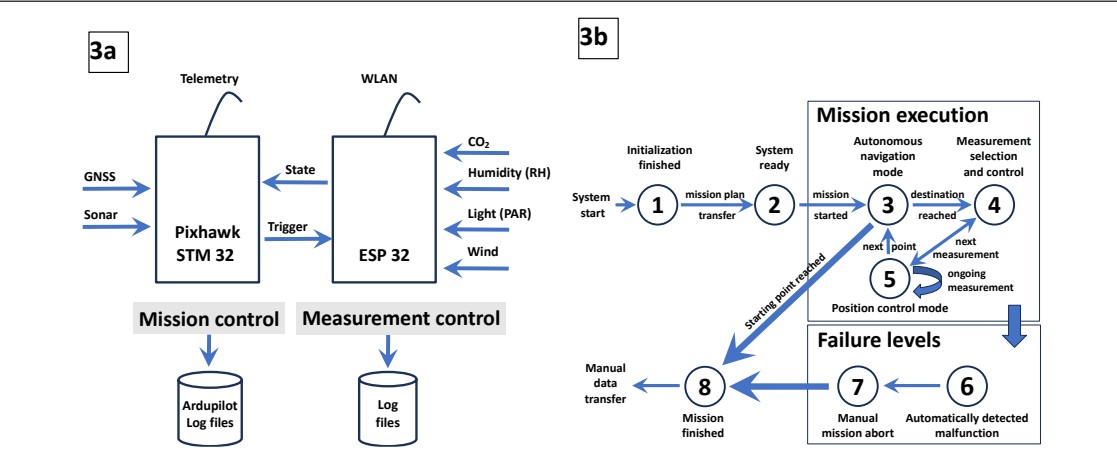

**Fig. 3a.** Robotic platform architecture. L: Components required for autonomous navigation (Mission control).
R: Integration of the actual mission sensors (Measurement control). **Fig. 3b.** Mission steps as state machine. The
implementation of a measurement task is sequenced in eight states (1–8) that can be divided into three levels:
initialization, mission execution and failure. States 3–5 are central. Here, the robot realizes the actual measurement
task and navigates between positions or executes changing programs at individual positions






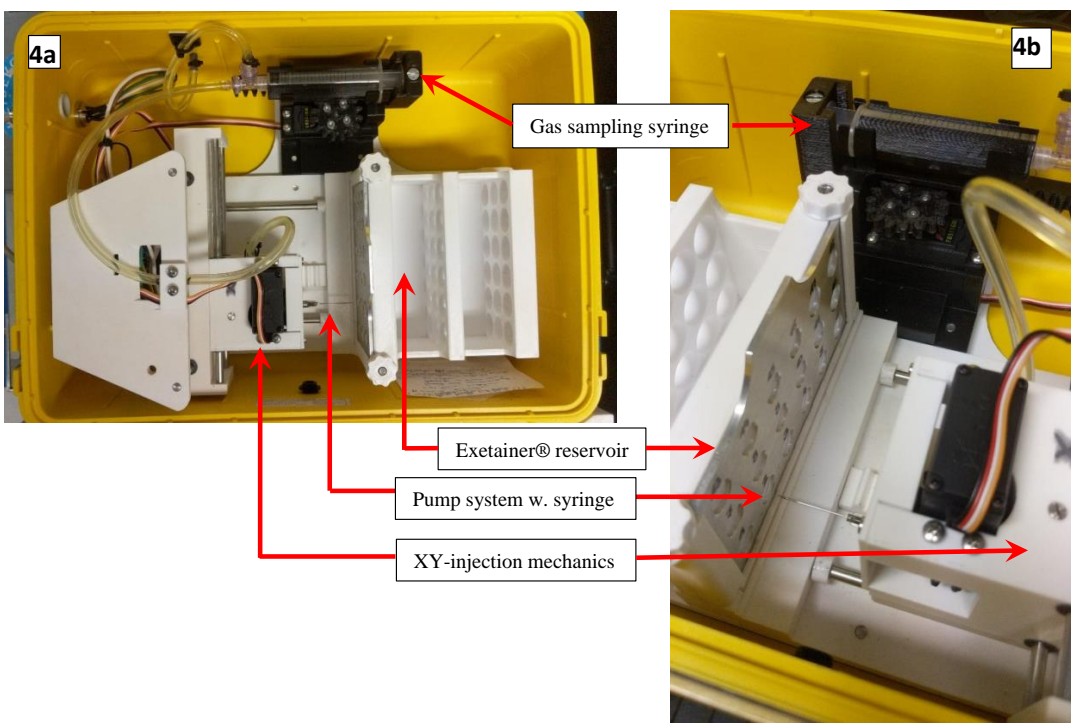

**Fig. 4.** Overview of the gas sampling unit (a) and details (b). **a)** The three central components: Pump system (10 mL) in the upper part, the autonomously operating injection mechanism and the reservoir for 3x6 Exetainers® on the right. **b)** The injection needle (at front) through which gas is being filled into the evacuated Exetainers® (left).



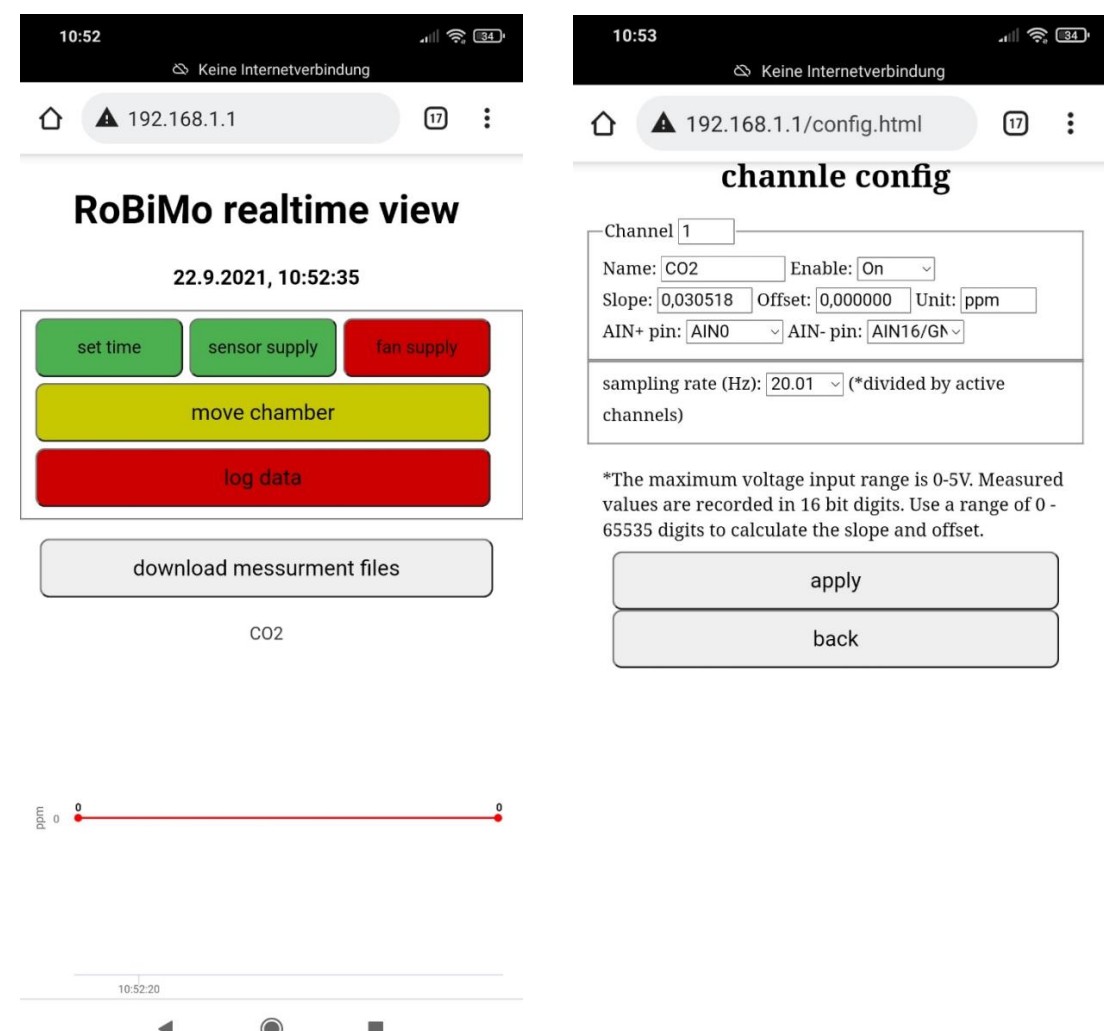

**Fig. 5.** Exemplary screenshots of the Website provided by the measurement system. It controls configuration parameters and visualizes the ongoing data aggregation in a diagram.




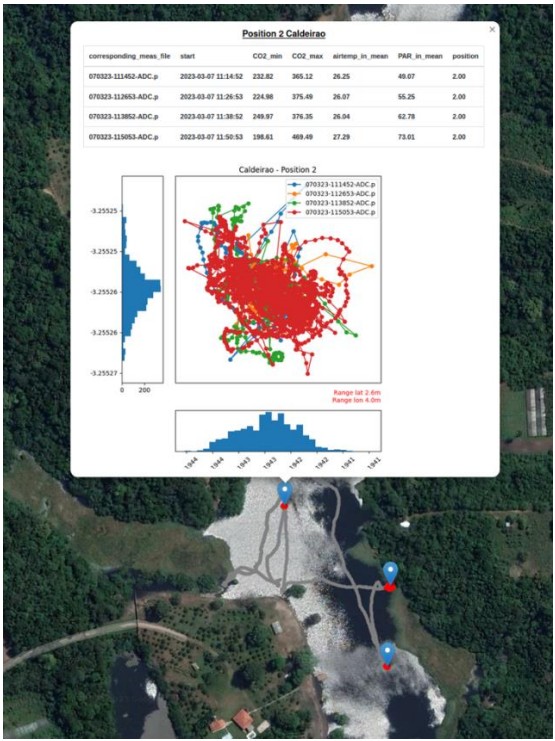


**Fig. 6.** MARP-FG tracks (grey) between sampling positions on Lake Caldeirão, Amazonas, Brazil. The inset shows the
platform movement at a specific position while performing measurements (4[th] RoBiMo/Trop campaign with minor flaws
in autonomous-cruising capability). During measurements the position of the system varies in an area of 4 x 2.6 m. The
histograms for latitude and longitude illustrate that the maximum deviation is caused by outliers

607





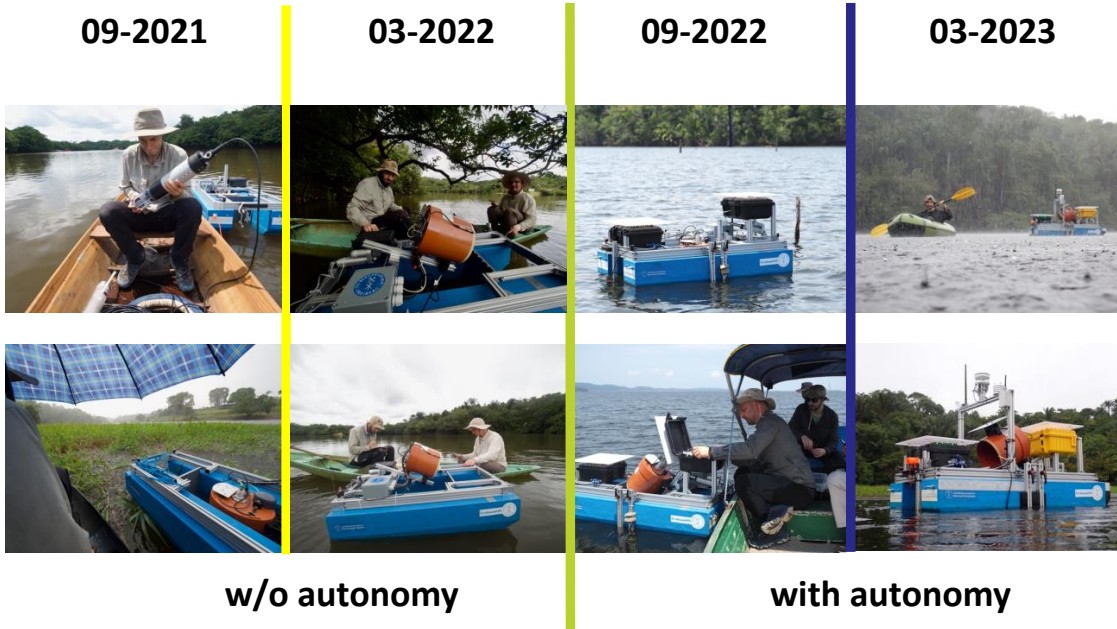

**Fig. 7.** Platform development from September 2021 to March 2023. Subsequent progress until August 2023 is invisible (= software improvements)