# Peer review of "Taking the pulse of nature – How robotics and sensors assist in lake and reservoir management"

_EGUsphere, 2024_

## Author Comment (AC1)

**Additonal replies to the referees comments**

We thank both referees for their critical assessment of our manuscript and are happy to reply to the additional questions raised. The arguments raised by the referees are repeated here with bullet points, and our related replies in italics to allow for a rapid overview. Given that this is the first time that we submit to "Geoscientific Instrumentation Methods and Data Systems", we hope not to overlook any formal requirements related to our communications. Kindly make us aware, if other or additional steps are to be taken.

- Clarify the comparison between MARP-FG and other robotic platforms in environmental monitoring to emphasize unique contributions.

  *In comparison with other robotic platforms, MARP-FG seems to be the most versatile with the ability to very quickly exchange payloads, lightweight and easy to transport. MARP-FG floater construction is the only one that generates calm water conditions between the floaters, allowing for high-quality gas exchange measurements and gas sampling.*

- Provide more quantitative results on data quality improvements achieved with MARP-FG (e.g., stability metrics in greenhouse gas measurements compared to manual methods).

  *The biggest advantage lies in the ability to perform mesaurements during nighttime and under adverse weather conditions, such as strong winds and rainstorms. Both are next to impossible without the platform.*

  *A PhD dissertation by Eric Roeder analyzed (among other issues) how gas sampling affects the in situ CO2 measurement. Sudden air pressure peak values occur with manual gas sampling, but not with automatic gas sampling. No noticeable or quantifiable influence were detected on the in situ CO2 measurement. We can deliver related material if needed.*

  *Manual sampling creates artificial turbulence that enhances gas exchange under calm water conditions. In Brazil, turbulence was no relevant factor due to the high emission rates, but in general this is important for GHG measurements: The less artificial turbulence, the better. Corresponding literature is available upon request.*

- Discuss any potential environmental impacts of the platform itself, such as energy use and interactions with local wildlife. How does MARP-FG minimize its impact during deployment in sensitive ecosystems?

  *The platform seems not to disturb wildlife at all. We observe e.g., curious freshwater dolphins checking the platform out then loosing interest and leaving again. We never observed waterfowl being interested in the platform. The platform is small and light, we have never observed any adverse effect.*

  *There are no emission sources coming from the platform, given its electical thrusters. We do not underdstand the question "such as eneryg use",*

  *Deployment of a small robotic platform avoids sending humans out with their smell, noises, and movements. There is much less water turbulence and no larger boat which may also mechanically disturb floating plant communities.*

- Expand details on MARP-FG's modular adaptability in different missions, as this is a core strength.

  *Within a few minutes, any payload can be detached and another payload mounted and activated. Quick release knobs hold any payload frame tight on the platform frame. The only steps needed are the detachment of cables and other connectors between platform and payload, then release the knobs and take th ecomplete payload off. Place another payload on, tighten the knobs and attach all necessary cabels and/or connectors. Ready.*

- Given the energy-intensive tasks (e.g., 3D sonar mapping), detailing power efficiency strategies, battery hot-swapping procedures, or even potential renewable energy options (such as modular solar panels) could enhance the platform's operational range and environmental sustainability.

  *The battery management is programmed for maximum efficiency, yet uses standard (off the shelf) components. The batteries are standards ones from cordless power tools, easily available almost everywhere. Any battery can be hot-swapped.*

  *Solar panels could be mounted. Their role would be to recharge the batteries. We did not need that option which would add to wind resistance, etc., as 8 hrs operating time plus hot swapping were fully sufficient and recvharging batteries on*

  *For the 3D mapping task, we now use another set of floaters, which are considerably longer (almost 2 meters) and can carry more weight (downside: Not that easy to transport when it comes to air transport). That set-up is powered by Minn Kota engines and much larger batteries (automobile batteries). However, the conventional MARP-FG setup, as described in the article can perform the task, too, albeit with lower speed and less hours.*

- You could provide a more discussion on MARP-FG's limitations in handling extreme environments. For instance, specifying the upper limits for humidity, temperature, or water turbulence where MARP-FG can still function optimally could help clarify its resilience.

  *There are no upper limits for environmental humidity or temperature, water wave heights up to ±40 cm and wind speeds to 7 meters per second still allow working the platform.*

- To improve data consistency across different mission types, establishing standardized data formats and describing these in the paper would aid researchers in efficiently managing and analyzing data from different payloads.

  *The status data of the robot is being recorded in the ArduPilot log format (*https://ardupilot.org/copter/docs/common-logs.html*). This format is supported by a variety of tools for analysis, debugging, and replay, making it well-suited as a standardized exchange format for use among researchers. In contrast, the format for individual gas exchange measurements and related contextual information is less standardized. These data are stored in CSV files, with the sensor characteristics documented in the corresponding publications.*

- Some of the hyperlinks in the paper are not functional.

  *There are nine (9) hotlink in the manuscript, see below. We checked everyone of them and they all worked well. We will check it again when the proofs come in, since sometimes, glitches may occur with typesetting or file transformation.*

    *https://ardupilot.org/planner/:*

> *https://sebastianzug.github.io/RoBiMo_Trop_DataSet/*
>
> *https://ardupilot.org/copter/docs/logmessages.html*
>
> *https://ardupilot.org/copter/docs/common-downloading-and-analyzing-data-logs-in-mission-planner.html#common-downloading-and-analyzing-data-logs-in-mission-planner*
>
> *https://gitlab.rrz.uni-hamburg.de/bay2789/bslogfiles/-/tree/master*
>
> *https://sebastianzug.github.io/RoBiMo_Trop_DataSet/html/balbina.html*
>
> *https://sebastianzug.github.io/RoBiMo_Trop_DataSet/html/interactive_table.html*
>
> *https://github.com/nteract/papermill*
>
> *https://www.pangaea.de/*

- On page 9, line 367, there's an incorrect reference to a figure. The text is explaining Figure 6, but it mistakenly says Figure 5.

  *You are perfectly correct, this is a mistake and has been corrected.*

**Questions:**

- Why was there no detailed statistical analysis or interpretation of the greenhouse gas flux data presented, especially given the claimed advantages of continuous data collection by the robotic platform?

  *This was expressed as overloading the paper by the editors. We are preparing another paper, solely dedicated to the respiration data. Much of the data will already appear in a PhD dissertation by Eric Roeder from TU Bergakademie Freiberg to be defended shortly. At the same time, an upload of the original data with all metadata in Pangaea has been done; publication is pending.*

- Given that the paper focuses on the reliability and accuracy of robotic deployment, what additional analyses could be conducted to demonstrate the environmental implications of the collected measurements, such as correlations between gas fluxes and environmental conditions?

  *One aspect has been addressed, namely wind speed. Other conditions include (heavy) precipitation and mixis.*

- What specific limitations or uncertainties exist in the gathered greenhouse gas data, and how could future studies leverage data analysis techniques to provide clearer environmental or biogeochemical conclusions?

  *So far, we have gained greenhouse gas data in five campaigns in both dry and rainy seasons in the Amazon basin. However, only the last two campaigns allowed for nighttime determinations. It would be highly desirable to dedicate more time to run more extensive experiments and data acquisition. Independent of the Amazon basin, the platform has been used successfully on various lakes and reservoirs in Central Europe.*

**Additional questions:**

- How do you plan to address limitations in MARP-FG's error communication and mitigation strategies for nighttime and long-distance missions?

  *In areas with mobile phone network coverage, data transmission occurs independently of the telemetry radio connection. This capability also enables the transmission of video*

*streams from the MARP-FG, facilitating detailed observation of platform operations, particularly during the development and testing of new functionalities. In the absence of this redundant communication channel, we realized extensive testing campaigns and developed a robust software system to ensure the robot's ability to operate autonomously. This design approach minimizes the reliance of the MARP-FG robot on live radio communication.*

- How does MARP-FG's design account for climate-induced changes, such as fluctuating water levels or extreme weather patterns? Have stress tests been conducted to ensure performance in these variable conditions?

  *See answer above (You could provide a more discussion on MARP-FG's limitations in handling extreme environments).*

- How is long-term data storage handled, especially for multi-season campaigns in remote areas? What strategies are in place to ensure data continuity and accessibility over extended periods?

  *A subset of the robot's filtered status data is freely available as a downloadable dataset on GitHub. The link is referenced in the paper. The dataset can be visualized and analyzed online through an interactive dashboard designed for interessted researchers. Additionally, the visualization includes a note indicating that the complete raw dataset can be provided upon request.*

- Could multiple MARP-FG platforms work together in coordinated tasks to cover larger areas more efficiently? If so, what communication protocols would be necessary to facilitate this collaboration?

  *Given the vastness of aquatic environments, the distribution of observed phenomena, and the dynamic nature of environmental conditions, deploying a swarm of MARP-FG systems could enable parallelized operations and facilitate synchronous measurements. For this approach, reliable communication between the robots would be essential to coordinate and trigger measurement processes effectively. A purely time-controlled execution of a pre-calculated plan would be inadequate due to the unpredictable jitter of travel times between measurement locations. Consequently, a mobile network infrastructure.*

---

## Author Response (AR2)

Dr. Jörg Matschullat
The Arthur L. Irving Institute, Dartmouth College
33 Tuck Mall Drive, Hanover, NH 03755, USA
joerg.matschullat@dartmouth.edu

**Replies to referees and editors (May 13, 2025)**

Dear Jean Dumoulin, dear anonymous referees,

We thank you kindly for your constructive criticism. Below, you find the latest conversation with our replies (italics). With uploading this letter comes the updated manuscript with the stated changes.

**Public justification (visible to the public if the article is accepted and published)**:

This paper presents a "ground-based" light and autonomous automated monitoring system for lake and reservoir management. The instrumented platform is clearly described and documented. Example of results obtained during a Brazilian experimental field campaign are completing the presentation of this autonomous aquatic instrumentation system. Few additional information would be appreciated to better illustrate the performances reached in term of navigation and $CO_2$ measurements.

*We thank the referees and our handling editor Jean Dumoulin explicitly for their constructive and encouraging comments. The remaining open questions regarding navigation performance (accuracy) and $CO_2$ determinations have been taken care of.*

**Additional private note (visible to authors and reviewers only):**

I first want to thank authors for their interesting paper addressing "ground based" autonomous floating robot instrumented system for in-situ monitoring of water lake and reservoir in good and harsh conditions. Concerning your autonomous navigation system versus in action planned navigation campaign:

*Thank you for your positive feedback. Your remaining questions have been taken care of in the revised manuscript. In addition, we quickly respond here.*

Did you make reproducibility test on a lake close to your laboratory? If yes, please add results obtained during such campaign? If, no I would like to recommend to add in your paper an overview of results obtained for the 6 Brazilian lakes experimental campaign. It will give a better overview of actual performances of your present system and may pave the way to possible enhancement.

*Yes, we did. Prior to the platform being taken to Brazil, we first tested ist development stages on a pond next to the institute and subsequently on ten different large reservoirs and pit lakes with different trophic levels – already in the context of another research project. In respect to results, we add a quote to one of our recent related publications (Röder E, Matschullat J, Rau A, Lau MP. 2024. Carbon dioxide emissions from temperate reservoirs and pit lakes of different trophic states. Inland Waters 1–45. doi 10.1080/20442041.2024.2388339).*

Why not using an additional local ground based station for your GNSS to work in differential mode ?

*An RTK solution is possible with two options: i) a physical reference station locally installed or ii) receiving data from an existing one in Brazil via the Internet. Option i): Here, we would have to add a radio link to connect the station and robot. Related to the context – the limited height of both systems' antennas and firm water surface damping effects – this is a challenging task. Option ii): This solution would always require an internet connection, likely unavailable in many scenarios.*

*In line with our approach to reducing complexity, we chose not to rely on an RTK reference station. Our focus was on developing a robust and self-contained system that performs reliably without depending on external infrastructure. The selected components are widely available and can be sourced in many countries worldwide during a campaign, ensuring operational flexibility and maintainability in diverse environments.*

Dr. Jörg Matschullat
The Arthur L. Irving Institute, Dartmouth College
33 Tuck Mall Drive, Hanover, NH 03755, USA
joerg.matschullat@dartmouth.edu

Concerning $CO_2$ measurements: Please give at least results comparison between in-situ measurements and laboratory measurements made on samples acquired on one lake. It should contribute to the underline vision of having less precision measurement with very dense spatial coverage combined with high precision measurements in few locations for data analysis and models.

*Generally speaking, each lake basin is an individual water body with different basin morphology, different trophic boundary conditions, etc. In addition, each basin is at least somewhat differentiated in itself. This means that there is no one gas respiration response for any basin. Depending on the scientific interest, one might set up a water-based eddy-covariance station to determine the respiration response over a larger lake water surface (e.g., Spank et al. 2023. Mobile eddy covariance measurements as a key to improve estimates of momentum, mass and energy fluxes between atmosphere and inland waters. PPNW 2023 Workshop on Physical Processes in Natural Waters, Brescia, Italy, 19.-23.06.2023; p 51-52). However that is very costly and makes sense only for longer-term monitoring purposes. If only shorter term and determinations that do not depend on air mass movement are wanted, then an aquatic robotic platform-based solution is superior and more economic Land-based eddy covariance towers are inadequate in this context, as they inevitably record mixed signals from both terrestrial and aquatic surfaces.*

*Current satellite-based methods lack sufficient spatial resolution to provide reliable signals. gas flux measurements. While other airborne techniques may address these spatial limitations, they generally involve substantially higher costs.*

*The following figure illustrates general observations comparing in-situ (on lake) $CO_2$-measurements with laboratory $CO_2$-determinations based on the gas samples collected concurrently with the on-board quantification using the Vaisala sensor.*

[Figure]

**Figure 1** *Comparison of in situ measurement (Vaisala sensor, solid line) and gas sampling (gas chromatography, open circles) for an example $CO_2$ time series*

*As the figure shows, the Vaisala sensor tends to deliver systematically lower mixing ratios as compared to the gas-chromatographic determination of the gas sampled obtained in parallel. This does not affect the flux calculation however, since the curves show rather similar behavior. We generally use the results from both methods in analyzing and interpreting the data.*

*Looking forward to see our work made available to a wider audience, we are glad that our work found appreciation.*

*Sincerely and for all authors and co-authors*

Dr. Jörg Matschullat
The Arthur L. Irving Institute, Dartmouth College
33 Tuck Mall Drive, Hanover, NH 03755, USA
joerg.matschullat@dartmouth.edu

*Jörg Matschullat and Sebastian Zug*